# Emergence of Cancer-Associated Fibroblasts as an Indispensable Cellular Player in Bone Metastasis Process

**DOI:** 10.3390/cancers12102896

**Published:** 2020-10-09

**Authors:** Naofumi Mukaida, Di Zhang, So-ichiro Sasaki

**Affiliations:** Division of Molecular Bioregulation, Cancer Research Institute, Kanazawa University, Kakuma-machi, Kanazawa, Ishikawa 920-1192, Japan; zddione@stu.kanazawa-u.ac.jp (D.Z.); s_sasaki@staff.kanazawa-u.ac.jp (S.S.)

**Keywords:** bone metastasis, cancer-associated fibroblast, fibroblast, mesenchymal stem cell, myofibroblast

## Abstract

**Simple Summary:**

Bone metastasis is frequently complicated in patients with advanced solid cancers such as breast, prostate and lung cancers, and impairs their prognosis. Bone metastasis proceeds through the interaction between cancer cells and resident cells in bone. Among resident cells, osteoclasts are commonly activated in bone metastasis, and therefore, the drugs targeting osteoclast activation are frequently used to treat bone metastasis. However, their ineffectiveness to inhibit cancer cell growth in bone marrow, raises the possibility of the involvement of additional types of resident cells in bone metastasis. Cancer-associated fibroblasts (CAFs) are fibroblasts that accumulate in cancer tissues as well as metastatic organs including bone. Hence, we will discuss the potential roles of CAFs, which are emerging as an important cellular player in bone metastasis.

**Abstract:**

Bone metastasis is frequently complicated in patients with advanced solid cancers such as breast, prostate and lung cancers, and impairs patients’ quality of life and prognosis. At the first step of bone metastasis, cancer cells adhere to the endothelium in bone marrow and survive in a dormant state by utilizing hematopoietic niches present therein. Once a dormant stage is disturbed, cancer cells grow through the interaction with various bone marrow resident cells, particularly osteoclasts and osteoblasts. Consequently, osteoclast activation is a hallmark of bone metastasis. As a consequence, the drugs targeting osteoclast activation are frequently used to treat bone metastasis but are not effective to inhibit cancer cell growth in bone marrow. Thus, additional types of resident cells are presumed to contribute to cancer cell growth in bone metastasis sites. Cancer-associated fibroblasts (CAFs) are fibroblasts that accumulate in cancer tissues and can have diverse roles in cancer progression and metastasis. Given the presence of CAFs in bone metastasis sites, CAFs are emerging as an important cellular player in bone metastasis. Hence, in this review, we will discuss the potential roles of CAFs in tumor progression, particularly bone metastasis.

## 1. Introduction

Various types of solid tumors, particularly breast, prostate and lung cancer, frequently metastasize to bone during their course and the bone is the third commonest site of metastasis after the liver and lung [1]. Metastasis proceeds through multiple steps, which consist of growth at the primary site, intravasation into the systemic circulation, dissemination via circulation, extravasation into the metastatic organ, and growth at the metastatic organ [2]. At the final step of metastasis, the metastatic cancer cells should survive and grow at the metastatic organ by utilizing and adapting its microenvironment, which is totally different from that of their primary site. Similarly, in order to facilitate bone metastasis, cancer cells should exploit and modify the bone marrow microenvironment, which consists of a myriad of cell types including hematopoietic cells, endothelial cells, osteoclasts, osteoblasts, mesenchymal stromal cells (MSCs), and fibroblasts [3].

The tumor microenvironment morphologically resembles a healing wound site [4], which is composed of multiple histological changes: extravascular clotting, inflammatory cell infiltration, angiogenesis, accumulation of activated fibroblasts, and synthesis of extracellular matrix (ECM) [4,5]. Activated fibroblasts in wound healing sites, display a contractile phenotype with enhanced expression of α-smooth muscle actin (α-SMA) and therefore are referred to as myofibroblasts. They produce ECM, various growth factors, cytokines, and chemokines, thereby promoting wound healing. Fibroblasts present in tumor tissues are denoted as cancer-associated fibroblasts (CAFs), and similarly display an activated phenotype with enhanced α-SMA expression. Like myofibroblasts in wound healing sites, CAFs can produce a wide variety of growth factors, cytokines, and chemokines. Based on these properties, it is widely accepted that CAFs have diverse and profound impacts on tumor growth, particularly at its primary site, but lack of a specific marker has hindered a clear-cut understanding of their roles, particularly at metastatic sites.

After summarizing the origin and functions of CAFs, we will discuss the roles of CAFs in bone metastasis, together with the incorporation of fibroblast-related cells and MSCs into consideration.

## 2. Phenotypes and Origins of Fibroblasts and CAFs

Fibroblasts were first described by Virchow as cells located in tissues that synthesize collagen and other ECM proteins [6]. These cells are present in connective tissues, as non-epithelial, non-vascular, non-leukocytic cells [7]. Embryonic fibroblasts appear in the primary mesenchyme which develops from epiblasts undergoing epithelial–mesenchymal transition (EMT) [8]. The primary mesenchyme transforms into the mesoendoderm and subsequently to the endodermal and mesodermal layers. The mesoderm yields to the true mesenchyme, which generates resident fibroblasts along with connective tissues [7]. Mesodermal layers generate several additional types of cells including endothelial cells, pericytes, adipocytes, and epithelial cells. Additionally, MSCs arise from mesoderm and are present in various adult tissues [9]. MSCs are defined as the plastic adherent cells, which can differentiate in vitro into osteoblasts, adipocytes, and chondrocytes, and exhibit expression of CD105, CD73, and CD90 but lack that of CD45, CD34, CD14 or CD11b, CD79a or CD19 and HLA-DR surface molecules [10]. Fibroblasts are presumed to be ubiquitously present in normal bone marrow [11] but they share phenotypic and functional similarities with MSCs [12] (Table 1), which are also abundantly present in bone marrow. Thus, it is frequently difficult to discriminate fibroblasts from MSCs.

Recent single-cell RNAseq analysis has revealed that fibroblast clusters could be discriminated from MSCs and pericytes due to high expression of fibroblast-specific genes including *fibronectin-1*, *S100a4*, *decorin*, and *semaphorin C* and low expression of chondrocyte-specific genes such as *Sox9*, *Acan*, and *Col2a1* [13]. The same group further revealed the presence of five distinct fibroblast clusters, fibroblast-1s to -5s, based on their gene expression signatures, suggesting their functional heterogeneities. Fibroblast-1s and fibroblast-2s cells expressed the progenitor markers *CD34* and MSC markers such as *Ly6a*, *Pdgfra*, *Th1*, and *Cd44*, but not endothelial- or pericyte-specific markers such as *Cdh1* and *Acta2*. Fibroblast-1s also expressed *Cxcl12* and *Angpt1* genes, which were abundantly expressed also by CAFs. On the contrary, fibroblast-3s, fibroblast-4s, and fibroblast-5s displayed gene expression of both *Sox9* and *scleraxis*, the transcription factors involved in the differentiation of tenocytes and ligamentocytes, while these three clusters formed a continuum. Moreover, fibroblast-4s and fibroblast-5s expressed bone- and cartilage-related genes including *Spp1* and chondrocyte genes such as *Nt5e*, *Cspg4*, amd *Clip*, compared with fibroblast-3s. Hence, it is probable that fibroblast-3s may be a tenocyte precursor and that fibroblasts-4s and fibroblast-5s may consist of tendon and/or ligament cells. AML-bearing mice exhibited changes in fibroblast cluster populations, compared with normal bone marrow: an increase in fibroblast-2 cluster and a decrease in fibroblast-5 cluster [13].

Resident fibroblasts usually lack the markers which are presumed to be restricted to other types of mesoderm-derived cells including hematopoietic cells (CD45), epithelial cells (E-cadherin), endothelial cells (Ve-cadherin, PECAM-1), pericytes (NG2), and adipocytes (uncoupling protein 1, UCP1) [7]. However, myofibroblasts exhibit activated phenotypes with enhanced contractility [14], and are characterized by expression of a set of relatively selective but non-specific markers, including α-SMA, fibroblast activation protein (FAP), fibroblast-specific protein (FSP)1/S100A4, THY1/CD90, platelet-derived growth factor receptor (PDGF-R)α/β, podoplanin, and discoidin domain receptor (DDR)2 [15]. Resident fibroblasts can yield to myofibroblasts under the influence of several fibrogenic factors, particularly transforming growth factor (TGF)-β family proteins [16] whereas other types of mesoderm-derived cells can transdifferentiate to yield to myofibroblasts [7] (Figure 1). Hence, myofibroblasts express hematopoietic, epithelial, endothelial, or pericyte markers to a variable degree, probably mirroring their origin.

CAFs are myofibroblast-like cells present in the tumor microenvironment and share phenotypic similarities with myofibroblasts in fibrotic tissues [17]. A substantial proportion of CAFs are presumed to be derived from resident fibroblasts [18] but can be originated from other types of cells [11,19,20,21], particularly mesoderm-derived cells, similarly to myofibroblasts (Figure 1).

TGF-β_1_ induced proliferating endothelial cells to undergo endothelial–mesenchymal transition (EndMT) to transform into CAF-like cells in the B16F10 melanoma model and the Rip-Tag2 spontaneous pancreatic carcinoma model [22]. EndMT was apparent at the invasive front of the tumors in the Rip-Tag2 spontaneous pancreatic carcinoma-bearing mice. Moreover, in the mouse lung tumorigenesis model, TGF-β_1_-mediated EndMT was accelerated when human pulmonary endothelial cells lacked an endothelial heat shock protein, HSPB1, which can protect against cellular stress [23], suggesting the involvement of cellular stress in this process. Another component of vasculature, pericytes, transdifferentiated into CAF-like cells upon PDGF-BB exposure and these CAF-like cells conferred a higher capacity to disseminate, invade, and metastasize on less-invasive cancer cells when co-injected [24]. Human normal mammary gland epithelial cells, HBFL-1 cells, acquired CAF-like phenotypes through EMT and bestowed marked increased tumorigenicity on a human breast cancer cell line, MCF-7, when being co-injected [25]. In addition to normal epithelial cells, some malignant epithelial cells also can undergo EMT to display CAF-like phenotypes [26]. Bone marrow-derived cells can also be transdifferentiated to display CAF-like phenotypes. Human MSCs were transformed to exhibit CAF-like phenotypes upon prolonged exposure to conditioned medium from human breast cancer cells and eventually promoted tumorigenesis when co-injected into animals with cancer cells [27]. Moreover, MSC-derived CAFs in breast cancer tissues exhibited reduced PDGF-Rα expression and augmented angiogenic activities, compared with resident fibroblast-derived CAFs [28]. Another type of bone marrow-derived cells, fibrocytes, which express type I collagen protein as well as a hematopoietic marker, CD45 [29], exhibited CAF-like phenotypes in chemically-induced rat breast carcinogenesis model [30]. However, there remain controversies on the contribution of non-resident fibroblasts to the generation of CAFs in solid tumor tissues [18], due to the lack of specific markers to identify CAFs (Figure 1).

Until present, only sketchy analyses have been conducted on the phenotypic and transcriptional profiles of CAFs in bone metastasis foci. The transcription profiles were markedly different in mesenchymal cells obtained from bone metastasis sites of breast cancer, compared with CAFs obtained from the primary sites or mesenchymal cells obtained from metastasized lymph nodes [31]. Moreover, several genes such as *HECTD1*, *HNMT*, *LOX*, *MACH1*, and *USP1*, were expressed to a larger extent in bone marrow stromal cells. Immunohistochemical analysis on human breast cancer tissues further revealed distinct expression patterns of fibroblast markers in CAFs at different metastatic sites [32]: high expression levels of podoplanin, S100A4, and PDGF-Rα in bone metastasis sites, high expression levels of PDGF-Rβ in lung metastasis sites, and reduced expression levels of S100A4 and PDGF-Rα in liver metastasis sites. However, the cause of phenotypic differences still remains elusive among CAFs at different metastatic sites, and their pathophysiological relevance.

## 3. Bone Metastasis Process

Metastasis advances through multiple steps [33], all of which are affected by CAFs [17,19]. At first, cancer cells should survive and proliferate at their primary sites, together with the induction of angiogenesis and immune evasion. Subsequently, they invade the adjacent tissues, intravasate into systemic circulation such as blood and lymphatics, migrate through circulation, and extravasate through vascular walls into the parenchyma of the distant organs to seed. During these processes, two specific phenotypic changes in cancer cells, EMT [34] or cancer cell stemness [35], have a crucial role. These processes proceed in a similar manner irrespective of metastatic organs. Finally, the seeding cancer cells should survive and proliferate, by utilizing a microenvironment that is provided by the metastatic organ and is quite different from that of the primary organ.

Bone marrow conducts two fundamental and vital functions, maintenance of bone structure and hematopoiesis. In order to perform these tasks, bone marrow consists of functional units with specialized structures, niches, to control bone remodeling and hematopoiesis. Niches can be classified into two types, endosteal and central ones, which are denoted depending on their location in bone marrow and in both niches, endothelial cells, their associated pericytes, and MSCs participate in the control of hematopoietic stem (HSC) fate [36]. Endosteal niches exist near transitional zone vessels and promote HSC quiescence while central niches are localized mostly near sinusoids and arterioles and control HSC transport. Endosteal niches contain additionally osteoblasts and osteoclasts to control bone remodeling [37]. Osteoclasts are differentiated from HSCs to resorb bone, while osteoblasts are differentiated from MSCs to mineralize collagen to form the calcified matrix of bone [38]. Osteoblasts provide pre-osteoclasts with receptor activator of NF-κB (RANK) ligand (L) and macrophage (M)-colony stimulating factor (CSF) to promote their differentiation to osteoclasts [38]. Differentiated osteoclasts resorb bone to release growth factors, which consequently induce the differentiation of osteoblasts from MSCs. Osteoblasts differentiate terminally into osteocytes, which are embedded in bone, to sense mechanical stress and deliver signals to osteoblasts and osteoclasts, thereby participating in bone remodeling (Figure 2) [38]. Cancer cells, which colonize bone marrow, hijack these niches, for their survival and proliferation [39]. 

At the first step of bone metastasis, cancer cells preferentially adhere to endothelium in endosteal niches [40,41] and start to grow therein through the interaction with the resident cells, similarly as cancer cells metastasizing to other organs do [2]. However, bone metastasis frequently follows a peculiar clinical course compared with that of other organs. A seminal study by Shiozawa and his colleagues revealed that in a prostate cancer bone metastasis model, colonizing cancer cells competed with HSCs for occupancy of the HSC niche [42]. Similar observations were obtained on a breast cancer bone metastasis model [43]. Thus, colonizing cancer cells utilize HSC niches and consequently can remain in a dormant state characterized by cell cycle arrest at G0 phase for a long period in bone marrow as HSCs can. Dormancy in intraosseous cancer cells may be sustained by additional mechanisms. TGF family proteins such as bone morphogenic protein (BMP) 7 [44] and TGF-β2 [45] constrained intraosseous cancer cells in a dormant state by shifting the balance between p38 and Erk kinases. Cancer cells in bone marrow produced vascular endothelial growth factor (VEGF) but its angiogenic effects were canceled by thrombospondin 1, which was released by endothelial cells in niches, thereby resulting in decreased tumor formation [46]. Furthermore, cancer cell growth in bone marrow can be compromised by interferon (IFN)-mediated cytotoxic immune response by CD8^+^ T lymphocytes and natural killer cells [47]. These mechanisms together may result in frequently observed long-latency of bone metastasis [3] and their disruption may lead to the development of clinically evident bone metastasis.

Once a dormant state is disturbed, osteomimicry ensues, where cancer cells stimulate both osteoclasts and osteoblasts, thereby inducing aberrant bone remodeling [39] (Figure 2). Bone metastasis lesions can be osteolytic or osteoblastic, depending on whether osteoclasts or osteoblasts are predominantly activated [1]. Moreover, osteomimicry is accompanied by a so-called vicious cycle, which consists of a positive feedback loop between aberrant bone remodeling and cancer cell proliferation [48]. In this positive loop which is known as a vicious cycle, cancer cells can augment osteoclast generation and activation directly by producing osteoclastogenic factors including RANKL [49], tumor necrosis factor (TNF)-α, IL-1, IL-6, and IL-11 [50]. Simultaneously, cancer cells activate osteoblasts by releasing parathyroid hormone-like protein (PTHrP), Wnt-1, insulin-like growth factor (IGF)-1, and BMPs and/or using Jag1/Notch pathway, and activated osteoblasts further enhance osteoclast generation and activation by releasing RANKL [50,51]. Activated osteoclasts resorb bone matrix and eventually release various growth factors such as fibroblast growth factor (FGF) and IGF [38,51], thereby promoting cancer cell growth in bone marrow. These observations imply that osteoclast activation can be a hub of a vicious cycle and as a consequence, bone metastasis is treated with drugs targeting osteoclast activation: antibody against RANKL or bisphosphonates [52]. However, a recent study using intravital microscopy revealed that cancer cells in a bone cavity efficiently grew even when osteoclast activation was inhibited by bisphosphonates [53]. Thus, it is likely that an additional cell component can participate in cancer cell growth in bone metastasis sites in addition to the cells involved in a vicious cycle. As bone marrow contains fibroblasts abundantly, particularly under pathological conditions, fibroblasts are emerging as a cell component that is potentially involved in bone metastasis. Hence, we will discuss their potential roles in bone metastasis and related conditions in the next section.

## 4. CAFs in Bone Metastasis Formation

CAFs are presumed to be involved in multiple metastasis steps from cancer cell growth at primary sites, their invasion, intravasation, migration through systemic circulation and extravasation, their seeding to distant organs and subsequent proliferation therein. CAFs can generally promote metastasis by inducing cancer cell survival and proliferation at the primary site in a paracrine manner, through secreting a myriad of growth factors and cytokines including heparin-binding epidermal growth factor (EGF) [54], epiregulin [55], hepatocyte growth factor (HGF) [56], IGF [57], TGF-β [58], interleukin (IL)-6 [59], CXCL12 [60], and IL-17B [61] (Figure 3). In addition to these growth factors, CAFs provided cancer cells with lactate, an energy source involved crucially in Warburg effects [62], thereby promoting tumor growth.

Cancer cell proliferation is promoted by angiogenesis which CAFs can profoundly induce by releasing various angiogenic factors. CAFs produce the most potent angiogenic factor, VEGF, abundantly [63]. CAFs abundantly produce IL-6, particularly in the presence of colon cancer cells and the produced IL-6 enhanced VEGF production by CAFs, leading to angiogenesis [64]. CAFs produce another potent angiogenic factor, PDGF-C, particularly in a compensatory manner when VEGF activity was inhibited [65]. Thus, this production may account for frequently observed resistance to anti-VEGF therapy. CAF-derived HGF also promotes angiogenesis, vascular mimicry, and mosaic vessel formation via PI3K/AKT and ERK1/2 signaling in gastric cancer tissues [66]. CAFs in colon cancer tissues produce FGF-1 and FGF-3, which acted at FGFR4 in endothelial cells, thereby promoting angiogenesis [67].

Cancer cell survival requires evasion from the host immune response. CAFs produce several immunosuppressive factors such as prostaglandin E_2_ [68], TGF-β1 [69] and VEGF [70], thereby directly suppressing tumor immunity. Additionally, CAF-derived cytokines and chemokines can promote the immune-suppressive tumor microenvironment by recruiting and generating immune suppressive cells. CAFs produced IL-6 and GM-CSF, which together induced the infiltration of tumor-associated macrophages (TAMs) and their differentiation into M2-like phenotypes in mouse syngeneic colon carcinoma models, thereby suppressing tumor immunity [71]. Moreover, CAFs conferred PD-1 expression and eventually immunosuppressive activities on TAMs via CCL2 and CXCL12 [72]. CAFs obtained from lung squamous cancer tissues produce CCL2, and eventually promote the recruitment of CCR2-expressing monocytes and their subsequent polarization into myeloid-derived suppressor cell (MDSC) phenotype, with a capacity to suppress autologous CD8-positive T-cell proliferation and IFN-γ production [73]. CAF-derived IL-6 induces the polarization of tumor-infiltrating T cells to Th17 phenotypes, thereby inducing tumor-promoting inflammation [74]. In contrast to these immune-suppressive functions, the depletion of CAFs induced immunosuppression with increased intratumoral regulatory T cells in a mouse pancreatic cancer model [75], suggesting a potential immune-enhancing function of CAFs in a context-dependent manner. However, it remains elusive how immune functional differences can be associated with phenotypic differences.

CAFs can enhance migratory and/or invasive capacity on cancer cells by releasing soluble factors. CAF-derived CXCL12 enhances the migration and invasion capacity of breast cancer cells [76] and gastric cancer cells [77]. Cancer cell migration and invasion has been augmented by other CAF-derived factors such as HGF, FGF-2 [78], and IL-8 [79]. CAFs can regulate migratory and invasive capacity in a cell-to-cell contact-dependent manner. Force transmission is mediated by a heterophilic adhesion involving N-cadherin at the CAF membrane and E-cadherin at the cancer cell membrane [80]. When subjected to force, this adhesion triggered β-catenin recruitment and adhesion reinforcement depending on α-catenin/vinculin interaction, leading to directional cancer cell migration and invasion. CAFs induced apoptosis in gastric cancer cells in a contact-dependent manner by using the death receptor 4-caspase-8 pathway [81]. Apoptotic cancer cells released apoptotic vesicles, which stimulated the invasion of CAFs and the subsequent migration of surviving cancer cells. CAFs produced a fibronectin-rich ECM with anisotropic fiber orientation, which guided the prostate cancer cells to migrate and invade directionally [82]. Furthermore, protease- and force-mediated ECM remodeling, which depends on integrins α3 and α5, and Rho-mediated regulation of myosin light chain (MLC) activity in CAFs, facilitated the squamous cell carcinoma (SCC) cells to invade in a cluster [83]. CAFs in pancreatic cancer form annexin A6/LDL receptor-related protein 1/thrombospondin 1 (ANXA6/LRP1/TSP1) complex, which is released in the form of extracellular vesicles (EVs), thereby enhancing the migratory capacity of cancer cells, which ingests EVs [84].

An important feature required for metastatic cells is the acquisition of EMT. CAF-derived TGF-β1 induces EMT in cancer cells [85]. Moreover, in an autocrine manner through the miR-200s/miR221/DNMT3B regulatory pathway, TGF-β1 maintained the active state of CAFs with a capacity to produce CXCL12, which promoted breast cancer cell proliferation [86]. CAFs produced also IL-6, which further promoted EMT in cancer cells in collaboration with TGF-β1 and the analysis on human non-small cell lung cancer tissues unraveled that IL-6 expression in CAFs was an independent prognostic factor [87]. Several CAF-derived soluble factors, such as HGF, Wnt, and PDGF also contribute to EMT phenotype acquisition in cancer cells [88,89,90]. CAFs promote EMT by secreting microRNAs such as miR-409, in addition to these proteinaceous factors [91].

CAFs can confer stemness, another feature required for metastatic cells, by releasing soluble factors or cell-to-cell contact. CAF-derived HGF activates the Wnt pathway in adjacent colorectal cancer cells, which leads to their acquirement of the stemness [92]. Autophagic CAFs abundantly release high-mobility group box 1 (HMGB1), which acts on its specific receptor, TLR4, on luminal breast cancer cells, to enhance their stemness [93]. The interaction between netrin and its receptor, UNC5B, maintained cancer stemness through the crosstalk between cancer cells and CAFs [94].

CAFs regulate ECM deposition, remodeling, and cross-linking, thereby stiffening stroma [95]. The resultant rigid stroma can enhance cancer cell survival, growth, and migration, induce its EMT, enhance hypoxia and subsequent angiogenesis, and compromise tumor immunity [95,96]. ECM solidification activates glycolysis in CAFs, which provided aspartate to sustain cancer cell proliferation and simultaneously triggered glutamine metabolism in cancer cells, which consequently balanced the redox states of CAFs to accelerate ECM modeling [97]. 

CAFs can regulate premetastatic niche formation, cancer cell seeding to bone marrow and their growth therein (Figure 4). α-SMA^+^ CAFs at the primary site of triple-negative (TN) breast tumors abundantly produced CXCL12 and IGF-1, the molecules that can select for cancer cells with high Src activity [98]. These cancer cells were prone to metastasize to the CXCL12-rich microenvironment of the bone marrow, suggesting that CAFs at primary sites educate cancer cells to metastasize to bone. Bone marrow stromal cells including fibroblasts may be required for CXCL12-rich microenvironment formation. Prostate cancer cells at primary sites released exosomes containing pyruvate kinase M2, which enhanced CXCL12 production by bone marrow stromal cells in a hypoxic inducible factor (HIF)-1α-dependent manner, giving rise to pre-metastatic niche [99]. Similarly, in prostate cancer bone metastasis, cancer cell-derived IL-1β conferred an activated CAF marker, S100A4 on bone marrow stromal cells and the resultant CAFs supported the colonization of cells, which otherwise exhibited a less metastatic capacity [100], although the detailed molecular mechanisms are not elucidated in this study. Thus, the interaction between cancer cells and CAFs can select bone-tropic cancer cells and can foster the so-called pre-metastatic niche formation in bone marrow.

The modification of ECM can facilitate bone metastasis formation. Lysyl oxidase (LOX), which is abundantly expressed in stromal cells, particularly fibroblasts, is a copper-dependent amine oxidase that catalyzes a key enzymatic step in the crosslinking of collagen and elastin [101]. As a consequence, it modulates and stiffens ECM. In an estrogen receptor-negative breast cancer bone metastasis model, cancer cell-derived LOX induced ECM solidification and subsequently activated osteoclasts through NFATc1 activation independent of RANK ligand and disrupted normal bone homeostasis, thereby providing a pre-metastatic niche for circulating tumor cells to colonize in bone marrow [102]. As a CAF subpopulation, CD146^-^ one abundantly express LOX [103], CAFs may be able to promote bone metastasis by modulating ECM with the use of LOX.

Breast cancer cells migrate from the primary site to bone marrow together with MSCs, while MSC migration is dependent on osteopontin [104]. Migrated MSCs express characteristic features of CAFs, vimetin and α-SMA expression, in an osteopontin-dependent manner. Cancer cells in bone marrow express a markedly higher expression level of stemness markers such as Nanog, Oct4, Sox2 in the presence of MSC-derived CAFs and osteopontin. Enhanced stemness in cancer cells can augment bone metastasis formation. Breast cancer cell stemness was also maintained by exosomes containing miR-221, which was abundantly released by vimentin-expressing FAP^+^ fibroblasts in bone marrow [105]. Thus, MSCs and their successor, CAFs, can contribute to breast cancer bone metastasis by maintaining cancer cell stemness.

Multiple myeloma (MM) is a malignant proliferation of monoclonal plasma cells, which secrete monoclonal immunoglobulin [106]. Malignant plasma cells infiltrate bone marrow, causing osteolytic lesions. Three phenotypical distinct CAF populations, FSP-1^+^α-SMA^−^, FSP-1^+^α-SMA^+^ or FSP-1^−^α-SMA^+^ ones were totally increased in bone marrow of MM patients at diagnosis and relapse than that of MM patients in remission or patients with monoclonal gammopathy of undetermined significance [107]. Moreover, CAFs induce chemotaxis, adhesion, proliferation and apoptosis resistance in MM cells through CXCL12 production and cell-to-cell contact mediated by the interaction between fibronectin and integrin. The same group further proved that MM cells abundantly secreted exosomes containing WW and C2 domain containing 2 (WWC2), which induced CAFs to express miR-27b-3p and miR-214-3p [108]. Overexpression of these miRNAs conferred apoptosis resistance on CAFs by reducing PTEN and FBXW7 proteins in CAFs. These observations would indicate that MM cells and CAFs reciprocally sustain their proliferation in bone marrow and that MM pathology is accelerated by the vicious cycle between MM cells and CAFs in bone marrow. 

Podoplanin^+^S100A4^+^PDGF-Rα^+^ fibroblasts are present in bone metastasis sites of human breast cancer patients [32]. Consistently, we observed an increase in fibroblasts in mouse breast cancer bone metastasis model [109]. In this study, we established a bone-tropic breast cancer cell clone, 4T1.3, from a mouse triple-negative breast cancer (TNBC) cell line, 4T1.0. Upon its orthotopic injection into the mammary fat pad, this clone exhibited a higher ability to metastasize to bone, which was ascribed to its enhanced capacity to grow in a bone cavity. Of interest is that neither osteoclasts nor osteoblasts were markedly increased in bone metastasis sites. A subsequent analysis revealed that cancer cells in a bone cavity produced abundantly a chemokine, CCL4, which attracted type I collagen-expressing α-SMA^+^ fibroblasts expressing CCR5, a specific receptor for CCL4 [109]. Accumulated fibroblasts provided cancer cells with a growth signal mediated by connective tissue growth factor (CTGF).

Prostate cancer cells and α-SMA^+^ CAFs cooperatively recruited macrophages and polarized them to M2-like TAMs via CCL2, CXCL12, and IL-6, while recruited macrophages conferred malignant and activation phenotypes on cancer cells and CAFs, respectively [110]. The same group further observed that the administration of a bisphosphonate, zoledronic acid, reduced bone metastasis formation in a murine prostate cancer bone metastasis model, in association with the reversion of M2-macrophage-mediated activation of CAFs [111]. Another group reported the possibility of more complicated interaction among prostate cancer cells, fibroblasts, and macrophages in the bone metastasis process [112]. A heparin sulfate proteoglycan, perlecan, and its modifiers, SULF1, were abundantly present at α-SMA^+^ fibroblasts in prostate cancer bone metastasis sites. M2-macrophage-derived condition medium increased in vitro perlecan and SULF1 expression in α-SMA^+^ CAFs. Moreover, SULF1 reduced Wnt3a-induced prostate cancer cell growth in tri-culture system consisting of prostate cancer cells, fibroblasts, and macrophages [112], although further investigation is warranted to determine the clinical relevance of these observations.

α-SMA^+^ CAFs were consistently seen in the bone of human patients with oral squamous cell carcinoma (OSCC), particularly at the head of cancer cells [113]. In vitro co-culture of OSCC-derived cells with CAFs increased RANKL expression and reciprocally decreased osteoproteogerin expression. Thus, the interaction between CAFs and cancer cells induced osteoclastogenesis and eventually accelerated bone invasion by OSCC cells [113].

Bone pain is common in patients with bone metastasis and impairs markedly their quality of life [1]. Bone-tropic breast cancer cells expressed abundantly V-ATPase and MCT4, leading to intratumoral acidosis, which induced α-SMA^+^FAP^+^ CAFs, osteoblasts and MSCs to express inflammatory and nociceptive mediators including nerve growth factor, brain-derived neurotropic factor, IL-1β, IL-6, CXCL8, and CCL5 [114], which may be responsible for bone pain.

CAFs are generally presumed to be pro-tumorigenic and pro-metastatic. However, they can exert both pro- and anti-tumorigenic activities even in similar experimental situations, and these discrepancies may be explained by phenotypic differences among CAFs. For example, inflammation and subsequent colitis-induced carcinogenesis were suppressed and augmented by *IKKβ* gene deletion in intestinal type IV collagen-positive fibroblasts [115] and that in type I collagen-positive intestinal fibroblasts [116], respectively. Thus, functional differences may arise from the differences in the origin of CAFs [28]. Moreover, CAFs, which were located at the outer edge of the tumor and were in close contact with T cells, suppressed T cell proliferation in a nitric oxide-dependent manner, whereas those around vessels lacked immunosuppressive activities [117]. Thus, the differences in their intratumoral locations may determine their functions. Comprehensive gene analysis on CAFs in breast cancer tissues revealed the presence of distinct subpopulations, which changed over time in breast cancer progression [118], without further elucidation on their functional relevance. Nevertheless, the lack of specific surface markers has hindered detailed phenotypic and functional clarification on CAFs involved in each step of bone metastasis. Thus, the application of novel research tools, particularly single-cell RNAseq analysis, is warranted in order to have a comprehensive perspective on the roles of CAFs in carcinogenesis including bone metastasis.

## 5. Conclusions and Future Perspective

We herein discussed the potential contribution of CAFs in bone marrow to bone metastasis. However, the most puzzling problem with bone metastasis is its frequent long-latency [51] and the roles of CAFs in this process remain elusive. Tumor cells are commonly detected in the bone marrow of patients with early-stage ductal carcinoma in situ (DICS) of breast cancer [119] or pathologically localized prostate cancer [120]. Additionally, overt bone metastasis was seldom seen in several types of cancers, such as gastric, colorectal, pancreatic, and cervical cancer, which occasionally harbor tumor cells in bone marrow [121,122]. It is widely admitted, also, that a significant proportion of bone metastases develop very late, sometimes decades after the diagnosis of primary tumors [123], in contrast to metastases of other organs, which becomes clinically evident within a short time frame due to the rapid growth of metastatic cancer cells. Thus, it is tempting to speculate that the bone marrow niche, which can originally support the survival of quiescent HSCs, is hijacked by dormant tumor cells for their survival. Moreover, it is likely that bone metastasis formation requires the re-entry of dormant tumor cells to a cell cycle in response to some cues from the bone marrow microenvironment. In the mouse breast cancer lung metastasis model, type I collagen accumulation induced dormant tumor cells to re-enter into a cell cycle through the β1-integrin-mediated activation of SRC, focal adhesion kinase, and Erk in tumor cells [124]. Considering that fibrotic changes are frequently observed in clinically overt bone metastasis sites, CAFs can similarly trigger the re-entry of dormant tumor cells in bone marrow to a cell cycle. Thus, it may be reasonable to assume that the manipulation of intraosseous CAFs will lead to the development of a novel strategy to prevent and/or treat bone metastasis.

## Figures and Tables

**Figure 1 cancers-12-02896-f001:**
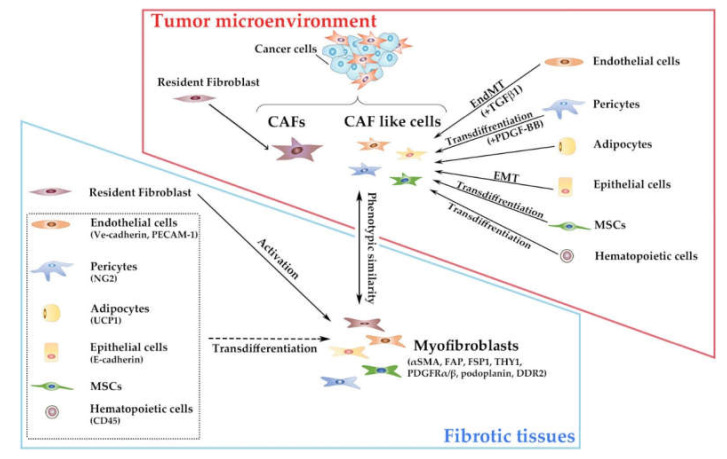
Ontogeny and phenotypes of cancer-associated fibroblasts (CAFs) and myofibroblasts. Parentheses indicate representative surface makers of each cell population. Used abbreviations: PECAM-1, platelet endothelial cell adhesion molecule-1; UCP1, uncoupling protein 1.

**Figure 2 cancers-12-02896-f002:**
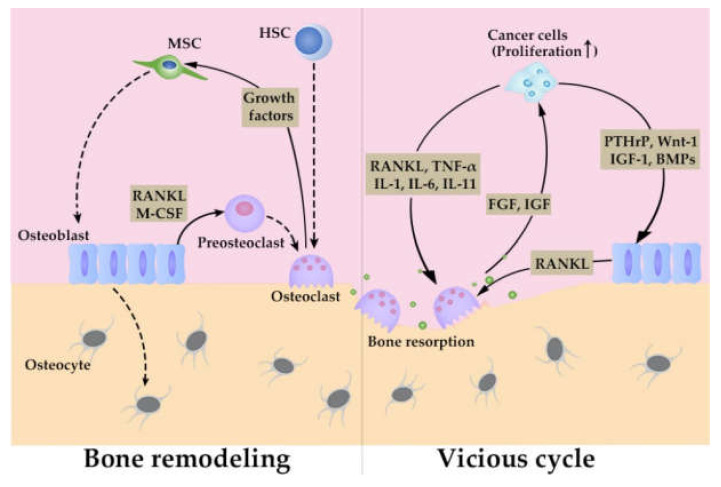
Bone remodeling and vicious cycle in bone metastasis.

**Figure 3 cancers-12-02896-f003:**
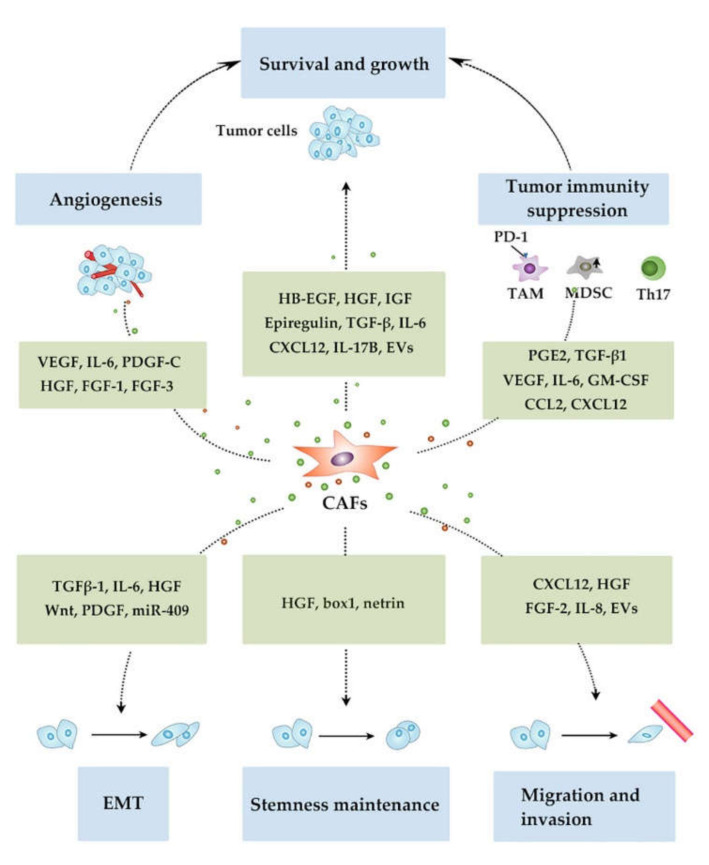
Functions of CAFs in tumor progression involved in metastasis.

**Figure 4 cancers-12-02896-f004:**
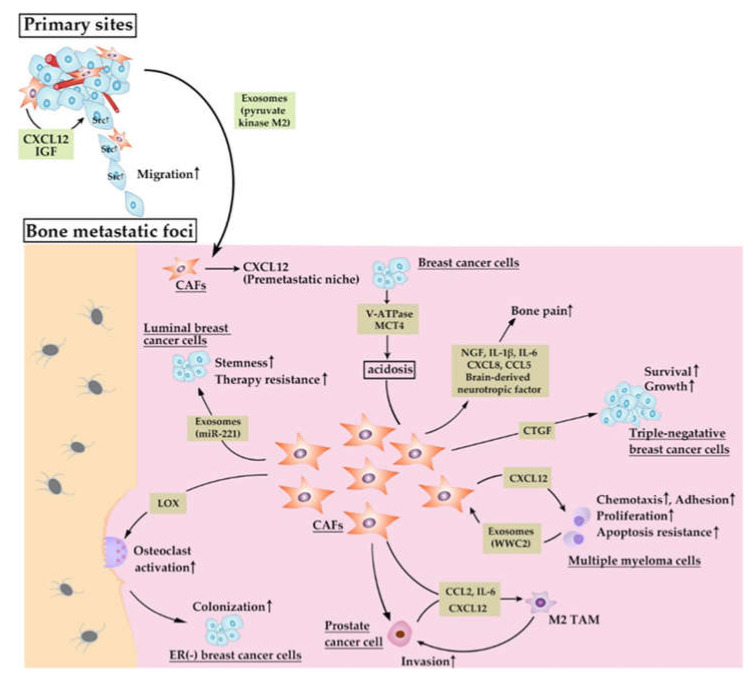
Presumed roles of CAFs in bone metastasis processes.

**Table 1 cancers-12-02896-t001:** Phenotypes of fibroblasts and mesenchymal stromal cells (MSCs). * phenotypes required to be identified as MSCs according to the proposal by The International Society for Cellular Therapy 2006 [12]. + positive; - negative; SSEA, stage-specific embryonic antigen.

Marker	Fibroblasts	MSCs
CD105	+	+ *
CD73	+	+ *
CD90	+	+ *
CD45	-	- *
CD34	-	- *
CD14	-	- *
CD19	-	- *
HLA-DR	-	- *
CD10	+/-	+/-
CD106	-	+/-
CD146	-	+/-
SSEA4	-	+
CD9	+	+/-
CD271	+/-	+/-
Stro-1	+/-	+/-

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
