# Peer review of "Emergence of Cancer-Associated Fibroblasts as an Indispensable Cellular Player in Bone Metastasis Process"

_cancers, 2020, doi:10.3390/cancers12102896_

Round 1

Reviewer 1 Report

Dear Authors

The topic is very interesting.

The article consist of six parts. The most important are: part 2 - Phenotypes and origins of fibroblasts and CAFs, part 3 - CAFs in tumor progression, part 4 - Bone metastasis process, part 5 - CAFs in bone metastasis and related conditions.

I have some comments, questions and suggestions for improvement.

  1. Keywords- they don't reflect the essence of the article. Authors should correct the keywords.
  2. The figures 1 and 2- The figures contain many abbreviations. To make it easier to understand the abbreviations should be explained in the description of the figure if it is possible. The tittle of the figure 1 - there is probably a small mistake - myelofibroblasts?
  3. The lines 53 and 54 - with this statement the primary source reference should be given.
  4. The line 112 - what does mean the abbreviation HBFL-1 cells?
  5. The lines 135-137 - does the reference 31 apply to all information in this sentence?
  6. The line 156 - what is the group box 1?

Author Response

The topic is very interesting.

The article consist of six parts. The most important are: part 2 - Phenotypes and origins of fibroblasts and CAFs, part 3 - CAFs in tumor progression, part 4 - Bone metastasis process, part 5 - CAFs in bone metastasis and related conditions.

I have some comments, questions and suggestions for improvement.

We appreciate the thoughtful comments from the reviewer. We indicate the major modifications in red. The followings are point-to-point response to the comments.

  1. Keywords- they don't reflect the essence of the article. Authors should correct the keywords.

In response to the recommendation, we changed the keywords as follows.

Bone metastasis, cancer-associated fibroblast; fibroblast, mesenchymal stem cell; myofibroblast

  1. The figures 1 and 2- The figures contain many abbreviations. To make it easier to understand the abbreviations should be explained in the description of the figure if it is possible. The tittle of the figure 1 - there is probably a small mistake - myelofibroblasts?

We explained the abbreviations used in the legends to figures, if they are not explained in the main text. We also changed the title per recommendation.

  1. The lines 53 and 54 (presently lines 53 and 54) - with this statement the primary source reference should be given.

In response to the recommendation, we replaced the reference 6 with the primary source.

  1. The line 112 (presently line 117) - what does mean the abbreviation HBFL-1 cells?

This is not an abbreviation but a name of cell line denoted by the authors of the original paper.

  1. The lines 135-137 (presently 136-140)- does the reference 31 apply to all information in this sentence?

The reference 31 (presently reference 32) apply to all information in this sentence.

  1. The line 156 (presently line 282) - what is the group box 1?

I am afraid that the reviewer may misunderstand and these words are a part of the term  “high-mobility group 1”. In order to avoid misunderstanding, we added the abbreviation of this word, “HMGB1”.

Reviewer 2 Report

Comment 1) Not well-focused on bone metastasis.

This manuscript consists of six chapters below.

1. Introduction , 2. Phenotypes and origins of fibroblasts and CAFs, 3. CAFs in tumor progression, 4. Bone metastasis process, 5. CAFs in bone metastasis and related conditions, and 6. Conclusion.

One chapter about bone metastasis, two chapters about CAF, and one chapter about CAF and bone metastasis.

Because most of the description of CAF, this manuscript would give the impression that the manuscript is focused on CAF.

Please summarize the sections about CAF and increase the sections CAFs in bone metastasis.

Comment 2) "Given the presence of CAFs in bone metastasis sites"

Many potential readers would believe the involvement of fibroblast but not CAF on bone metastasis.

This may be because many results indicating the involvement of CAFs in vitro, but few indicating involvement of CAF in vivo.

For example, desmoplastic reactions often observed in pancreatic cancers, but very rare in breast cancers.

The authors are encouraged to demnonstrate the "the presence of CAFs in bone metastasis" in vivo.

Comment 3) Which step of bone metastasis involves what phenotype of CAF?.

Although the authors demonstrated "Presumed roles of CAFs in bone metastasis processes" in Figure 4, this is not clearly described which step involves CAF.

Because fibroblasts are observed everywhere, it is possible CAF may be observed in every step of bone metastasis.

But CAFs may not always play important role in that step of bone metastasis.

Because many resports suggest that involvement of CAF on metastasis formation (non specific) and it would be interesting if the authors would describe the specific involvement of CAF on bone metastasis formation.

The authors are strongly encouraged to describe the role of CAFs on each step, such as leaving primary site, extravazation, re-growthg in the bone microenvironment, and interaction of bone stromal cells, separately.

These descriptions would help for the potential readers to understand what phenotype of CAF is involved in which step of bone metastasis, which would give us the new potential target of bone metastasis.

Author Response

We appreciate the thoughtful comments from the reviewer. We indicate the major modifications in red. The followings are point-to-point response to the comments.

Comment 1) Not well-focused on bone metastasis.

This manuscript consists of six chapters below.

  1. Introduction , 2. Phenotypes and origins of fibroblasts and CAFs, 3. CAFs in tumor progression, 4. Bone metastasis process, 5. CAFs in bone metastasis and related conditions, and 6. Conclusion.

One chapter about bone metastasis, two chapters about CAF, and one chapter about CAF and bone metastasis.

Because most of the description of CAF, this manuscript would give the impression that the manuscript is focused on CAF.

Please summarize the sections about CAF and increase the sections CAFs in bone metastasis.

In response to the recommendation, we re-organized the article by combining the previous section 3 with the previous section 5 to transform them into the new section 4. Accordingly, the previous sections 4 and 6 are presently sections 3 and 5, respectively. Moreover, we reduced the discussion in the previous section 3 and discussed the pro-tumorigenic actions of CAFs by interrelating with metastasis processes. Furthermore, we discussed the papers on bone marrow fibroblasts in bone metastasis in more detail.

Comment 2) "Given the presence of CAFs in bone metastasis sites"

Many potential readers would believe the involvement of fibroblast but not CAF on bone metastasis.

This may be because many results indicating the involvement of CAFs in vitro, but few indicating involvement of CAF in vivo.

For example, desmoplastic reactions often observed in pancreatic cancers, but very rare in breast cancers.

The authors are encouraged to demnonstrate the "the presence of CAFs in bone metastasis" in vivo.

We described the presence of fibroblasts in bone metastasis sites of human breast cancer (lines 338 to 339, reference 32) and mouse breast cancer (lines 339 to 340, reference 109). Moreover, we described explicitly that fibroblasts in bone have a crucial role in bone lesion progression in human multiple myeloma (lines 327 to 330, reference 107) and oral squamous cell cancer (lines 361 to 362, reference 113).

Comment 3) Which step of bone metastasis involves what phenotype of CAF?.

Although the authors demonstrated "Presumed roles of CAFs in bone metastasis processes" in Figure 4, this is not clearly described which step involves CAF.

Because fibroblasts are observed everywhere, it is possible CAF may be observed in every step of bone metastasis.

But CAFs may not always play important role in that step of bone metastasis.

Because many resports suggest that involvement of CAF on metastasis formation (non specific) and it would be interesting if the authors would describe the specific involvement of CAF on bone metastasis formation.

The authors are strongly encouraged to describe the role of CAFs on each step, such as leaving primary site, extravazation, re-growthg in the bone microenvironment, and interaction of bone stromal cells, separately.

These descriptions would help for the potential readers to understand what phenotype of CAF is involved in which step of bone metastasis, which would give us the new potential target of bone metastasis.

In response to the recommendation, we describe the roles of CAFs on each each step in the new section 4. Moreover, we describe the phenotypes of CAFs, which are reported to be directly involved in bone metastasis, from lines 293 to 370.

Round 2

Reviewer 2 Report

The authors responded all the concerns appropriately.

Author Response

The authors responded all the concerns appropriately.

I appreciate your highly favorable comments.